# Use of Lateral Flow Assays in Forensics

**DOI:** 10.3390/s23136201

**Published:** 2023-07-06

**Authors:** Brigitte Bruijns, Roald Tiggelaar, Jaap Knotter, Annemieke van Dam

**Affiliations:** 1Technologies for Criminal Investigations, Saxion University of Applied Sciences, M.H. Tromplaan 28, 7513 AB Enschede, The Netherlands; b.b.bruijns@saxion.nl (B.B.); j.c.knotter@saxion.nl (J.K.); 2Police Academy, Arnhemseweg 348, 7334 AC Apeldoorn, The Netherlands; 3NanoLab Cleanroom, MESA+ Institute, University of Twente, Drienerlolaan 5, 7500 AE Enschede, The Netherlands; r.m.tiggelaar@utwente.nl; 4Department of Biomedical Engineering & Physics, Amsterdam University Medical Centers, University of Amsterdam, Meibergdreef 9, 1105 AZ Amsterdam, The Netherlands; 5Department of Forensic Science, Amsterdam University of Applied Sciences, Tafelbergweg 51, 1105 BD Amsterdam, The Netherlands; 6Methodology Research Program, Amsterdam Public Health Research Institute, Amsterdam University Medical Centers (UMC), Location AMC, University of Amsterdam, Meibergdreef 9, 1105 AZ Amsterdam, The Netherlands

**Keywords:** lateral flow assays, forensic investigation, body fluid identification, illicit drugs analysis, explosives analysis

## Abstract

Already for some decades lateral flow assays (LFAs) are ‘common use’ devices in our daily life. Also, for forensic use LFAs are developed, such as for the analysis of illicit drugs and DNA, but also for the detection of explosives and body fluid identification. Despite their advantages, including ease-of-use, LFAs are not yet frequently applied at a crime scene. This review describes (academic) developments of LFAs for forensic applications, focusing on biological and chemical applications, whereby the main advantages and disadvantages of LFAs for the different forensic applications are summarized. Additionally, a critical review is provided, discussing why LFAs are not frequently applied within the forensic field and highlighting the steps that are needed to bring LFAs to the forensic market.

## 1. Introduction

Lateral flow assays (LFAs) are a class of paper-based microfluidic devices that are used to detect and quantify analytes in various complex mixtures. Paper is a biocompatible material, meaning that the material does not negatively influence the biological sample and/or reaction. Moreover, due to its capillary action using the liquid absorbing properties, there is no need for (external) equipment, such as pumps and valves, to move fluids along the device [1,2,3,4,5].

In the medical field, these LFAs are often used as point-of-care devices to aid as tool for rapid screening and monitoring of an individual’s health status. The most well-known examples of LFAs are the SARS-CoV-2 self-test [2] and the pregnancy test [6]. LFAs are also developed for the detection of *E. coli* [7], Zika virus [8], HIV [9] or nucleic acid detection in general [10,11]. Apart from these medical applications, LFAs have high potential for use within the forensic field, in which they are currently mainly used for illicit drug analysis [12,13] and body fluid identification [14].

However, within forensics LFAs are not fully integrated yet, and their application in this field can be extended. LFAs have many advantages over traditional laboratorial ana-lysis. LFAs are easy to handle, user-friendly and provide a quick indication of whether a certain target molecule is present in the sample of interest. The use of mobile identification methods directly at the crime scene—for example by the use of LFAs—is growing and the information extracted after analysis provides crime scene investigators first input to select and prioritize the evidence, but also to make fast decisions directly at the crime scene, and obtain investigation leads to test different scenarios [15]. A major drawback of LFA is that they provide qualitative and/or semi-quantitative results, therefore the obtained results can only be used as indicative tools. Additionally, some LFA tests show also cross-reaction with targets of interest with similar (chemical) structures, which might lead to false positive results. Quantitative confirmatory testing is needed to confirm the on-site LFA test results. In Figure 1 an overview is given of the advantages and the disadvantages of LFAs [3,16,17].

In this review, after introducing the principle of a LFA, a detailed overview is given of the recent developments of LFA use in the forensic field, with a separation between biological and chemical applications. Moreover, a critical reflection on the steps that are needed to speed-up the implementation of LFAs at the crime scene is provided.

## 2. Principle of Lateral Flow Assays

LFAs, schematically presented in Figure 2, are composed of a strip with often a plastic backing card. This strip consists of a sample pad, conjugate pad, flowing membrane and an absorbent pad, allowing the sample to travel along the paper using the capillary action in the (nitro)cellulose (NC) [18]. Depending on the sample type, the materials used for the sample, conjugate and absorbent pad should be selected carefully to guarantee optimal operation of the LFA for the specific utilization. Sample and absorbent pads are often comprised of cellulose, while glass fibers are used as conjugate pad. For analyte detection, as the first step, a sample needs to be collected. Sample preparation may be necessary, depending on the type of sample, which can range from a pure component diluted in a buffer to an analyte present in a complex sample matrix like blood or saliva. A dedicated (extraction) buffer is added to the sample for sample work-up. As the next step, the worked-up sample is introduced into the lateral flow device by dripping/dispensing a few drops onto the sample pad by the use of for instance a dropper bottle or a pipette. The sample migrates first through the conjugate pad (containing reagents) and then through the NC strip on which the actual reaction and detection take place. The conjugate pad contains dried reagents, such as antibodies, nucleic acids and/or enzymes, which are rehydrated upon the addition of the analyte solution. Analytes are able to react with the reagents present on the conjugate pad and will migrate further over the LFA toward the test and control lines. At this location, capture molecules are present that are able to react/bind to the analyte and will cause the complex of the detection molecule and analyte to bind on the test line of the LFA. A control line is often included to demonstrate that the LFA is functioning properly. In most cases, antibody-antigen reactions are used to capture and detect analytes of interest. However, also other capture molecules can be integrated in a LFA, such as aptamers, enzymes and DNA probes [19]. The detection molecule can be labeled with e.g., a fluorophore or nanoparticle for read-out by the unaided human eye (or in more complex cases a camera or smartphone) [11,20]. Fluorescence read-out methods are in general more sensitive, but require a read-out system [17]. 

Lateral flow assays have a high potential to be applied for forensic applications. In fact, such paper-based microfluidic devices are already used in forensics for biological (Section 3) as well as chemical (Section 4) applications: in the following (sub)sections examples are discussed.

## 3. Lateral Flow Assays for Biological Forensic Applications

In the following subsections, the use of LFAs for body fluid identification and DNA detection is reviewed. At the end, in Table 1 a summary is given of which LFA test(s) can detect which (classes of) analyte(s), including time-to-result and limit of detection (LOD). 

### 3.1. Body Fluid Identification

Within the forensic field, body fluid identification (BFI) is extremely important, as the origin of body fluid is indicative of the presence of DNA and additionally might help in reconstructing the events that took place at a crime scene, as well as for investigative leads [21,22]. Identification of body fluids is not an easy task, as their appearance can be similar to each other or other substances. Therefore, different presumptive and confirmatory methods are developed to identify human biological stains, including chemical, enzymatic, spectroscopic and microscopic methods [23,24,25]. LFAs can also be used as a rapid and specific tool to identify body fluid specific markers. The LFAs that are currently available for body fluid identification are mainly based on antibody-antigen interactions, whereby proteins are the target molecules. Protein-based interactions have the advantage of being highly selective and have high affinity with their target molecule. Several commercial LFA kits are available, such as the Rapid Stain Identification (RSID) tests from Independent Forensics, tests from SERATEC and BlueSTAR-forensics [26,27,28]. Most of these commercial tests focus on the identification of blood, semen, saliva and urine. These body fluids are frequently encountered at the crime scene, and specific biomarkers are identified that can be targeted using antibodies.

For the identification of blood, various commercial tests are available [29,30,31]. For example, the RSID kits are immunochromatographic assays, using labeled antibodies that react to body-fluid-specific markers. For example, the RSID blood test uses anti-glycophorin A monoclonal antibodies to specifically detect the blood specific glycophorin A marker, a protein that is expressed by red blood cells. With this test 0.05 µL of blood can be detected (note: a standard droplet of blood is around 50 µL). Importantly, no false-positive results were obtained when analyzing blood from other species, including other primate species, such as monkeys and baboons [30]. To discriminate between peripheral blood and menstrual blood, SERATEC developed a LFA that is able to detect two different biomarkers in one multiplex LFA: hemoglobin and D-dimer. Hemoglobin is present in peripheral and menstrual blood, whereas D-dimer is only present in menstrual blood. Since the D-dimer is not human-specific (a positive test result was obtained for rabbit blood samples as well), both lines should generate a positive signal to indicate the presence of human menstrual blood [32]. Apart from the SERATEC test, other commercial kits are also available that are able to discriminate menstrual blood from peripheral blood, including the Dade Dimertest Latex Assay, and OneStep D-dimer RapidCard InstaTest. Tsai et al. [26] evaluated these kits for their applicability to the forensic field of menstrual blood identification, as visualized in Figure 3. Their conclusion was that all three commercial kits could aid in the detection of menstrual blood. Differences in sensitivity were found when comparing the three kits: whereas up to 0.025 µL of dried menstrual blood stains had to be used to indicate the presence of menstrual blood for the Dade Dimertest Latex Assay and SERATEC assay, the OneStep D-dimer RapidCard InstaTest was able to positively indicate the presence of menstrual blood within a stain of 0.003125 µL.

For the detection of saliva, various commercial LFAs are available, including the RSID saliva test and SERATEC Amylase test [33]. In both tests, anti-amylase is included to test for the presence of amylase A, which is an enzyme that is produced by the salivary glands, and will break down carbohydrates to maltose. The test has a sensitivity of about 1 ng of α-amylase, which corresponds to 1 µL of saliva. No cross-reactivity is found with saliva samples from other species [34]. 

LFA tests are also developed to indicate the presence of semen. Many different commercial tests are available including the RSID semen test, ABAcard-p30, BLUESTAR Identi-PSA and the PSA Semiquant test from SERATEC. The RSID test detects the presence of semenogelin, a protein involved in the formation of a gel matrix enclosing ejaculated sperm cells [35,36], whereas the ABAcard-p30 and the PSA Semiquant test detect prostate specific antigen (PSA), also called p30. PSA is a serine protease that is produced by the seminal vesicles, liquefying the seminal clot. Both semenogelin and PSA are abundant markers present in seminal fluid, in concentrations of 4–68 mg/mL and 0.2–5.5 mg/mL, respectively [37]. The sensitivity of these rapid tests is high since they are able to detect semen diluted up to 1:50,000–1:100,000 [38,39]. In a study in which the RSID semen test was compared with the ABAcard-p30, minimal differences were found related to the sensitivity and the user-friendliness of the tests. However, samples obtained from vaginal swabs 24 h post-coital resulted in positive test results with the ABAcard-p30, whereas negative results were obtained with the RSID semen test [40]. Chang et al. [37] compared the performance of these three LFA tests and concluded that the RSID semen test is less specific than the PSA based LFA tests, since with the RSID semen test false-positive results were obtained when analyzing male urine, female urine and vaginal fluid. The incubation time of the sample with the extraction buffer for the RSID test is shorter, specifically 1 h for the RSID test versus 2 h for the ABAcard-P30 test. In a study by Kishbaugh et al. [41], the specificity of the SERATEC immunochromamotographic assays was further explored. Since the targets PSA and amylase are also found endogenously in the vagina, it was expected that the PSA and amylase tests should also respond positively to vaginal fluid. Indeed, in their study, 2% and 8% of the samples provided a positive test result with respectively the SERATEC PSA Semiquant test strip and the Amylase test. These results indicate that care should be taken when interpreting the results from these LFAs and that additional confirmatory tests are needed. 

To identify the presence of urine the RSID urine test is available, which detects the urine-marker Tamm-Horsfall protein. Positive test lines were obtained when urine was diluted 1:20. In the case of dried urine stains the identification can be difficult since the stain needs to be diluted in buffer, before it can be analyzed, which might result in a too diluted sample for analysis [42].

To date, no multiplex LFAs are commercially available, besides the SERATEC PMB test to discriminate menstrual blood from peripheral blood. A first step towards the development of a multiplex LFA for body fluid identification was taken by Holtkötter et al. [43], by combining LFAs from different manufacturers into one single assay for rapid detection (10–15 min) of up to five body fluids (with high specificity and sensitivity): semen, saliva, urine, menstrual blood and blood. Positive signals were obtained for semen, saliva and menstrual fluids using 10 nL of sample material, whereas for blood only 0.002 nL was required. No cross-reactivity was observed between the different body fluids and mixed samples. 

### 3.2. DNA

Another application of LFAs for forensic usage related to biological samples involves the detection of unique nucleic acid sequences. Upon hybridization of reporter and capture probes with the target sequence, the nucleic acid sequence of interest can be detected [44]. This technique can be used for the detection of, among others, spores and (human) DNA analysis. 

Anthrax spores can cause an infection, depending on contact and/or inhalation, to the skin, lungs and intestinal. Also, fever, chest pain and shortness of breath can occur after inhalation. These spores became of forensic interest after the spreading of anthrax letters in 2001, in which anthrax was used as a bioweapon. Hartley et al. [44] developed a lateral flow test for the detection of a mRNA sequence of the anthrax toxin activator gene, present in *B. anthracis* (anthrax) spores. After RNA extraction and nucleic acid sequence-based amplification (NASBA) of the RNA the sample is placed on the lateral flow assay. The capture and reporter probes are modified with a biotin at the 5′ end and an amine group at the 3′ end, respectively. The reporter probe is also coupled to a dye-encapsulating liposome of which the signal can be read out by the unaided eye or a (hand-held) reflectometer. 1.5 fmol of target mRNA could be detected. While the lateral flow assay itself only requires a reaction time of 15 min, the overall process took 12 h. Later on, the time-to-result was reduced to 4 h by changing the target sequence [44,45].

Apart from the detection of anthrax spores, the rapid identification of other highly dangerous pathogens is also of forensic interest. Zasada et al. [46] compared three different isothermal nucleic acids amplification methods, i.e., loop-mediated isothermal amplification (LAMP), thermophilic helicase-dependent isothermal amplification (tHDA) and recombinase polymerase amplification (RPA), to address their sensitivity when combined with lateral flow dipstick analysis. Three different pathogens, *B. anthracis*, *F. tularensis* (tularemia) and *Y. pestis* (plague) were included in this assay. Different reaction times were needed for the three DNA amplification methods, varying from 60, 90 and 30 min, respectively for LAMP, tHDA and RPA. All three methods could be combined with lateral flow dipstick analysis, resulting in positive detection of the pathogen of interest. However, the authors indicate that there is a risk of false-positive results in the case of pathogens with high genetic similarities to non-pathogenic species, which might be due to the isothermal amplification method and this should be further investigated. 

Prior to the lateral flow detection of male DNA Kubo et al. [47] used LAMP to amplify the target of interest. The time-to-result was around 60 min and 10 pg of male DNA could be detected. An assimilating probe and a biotin-labeled primer are used in the LAMP assay. The dual-labeled amplicons are subsequently captured, whereafter the fluorescent signal can be detected (Figure 4). 

Tungphatthong et al. [48] used PCR in combination with lateral flow analysis for the identification of a specific plant species, namely *M. Speciosa*, which has opioid-like properties and is classified as a narcotic drug in several countries. Three different labeled primers were included, which were tagged on the 5′ ends by different antigens to discriminate between the amplicons, and thus perform a multiplex analysis. The lateral flow assay showed good sensitivity, detecting as low as 10^−5^ ng of DNA. The method was able to discriminate *M. speciosa* from other *Mitragyna* species that exist in Thailand.

**Table 1 sensors-23-06201-t001:** Overview of characteristics of LFAs for body fluid identification and DNA detection.

Class of Analyte	Analyte(s) That Can Be Detected	Type of Test	LOD	Time-to-Result	Refs.
Blood	Glycophorin A	Lateral flow strip, RSID test	0.05 µL blood	Extraction: 50 min Analysis: 10 min	[29,30,31,49]
	Hemoglobin	Lateral flow strip, Seratec PMB test & Seratec HemDirect Hemoglobin test	20 ng/mL	Extraction: 10 min Analysis: 10 min	
Menstrual Blood	D-dimer	Lateral flow test, clearview rapid D-dimer test	1:30 diluted stain	Extraction: 60 min Analysis: 20 min	[26,32,49]
		Lateral flow test, PMB test, Seratec	400 ng/mL 0.025 µL dried stain	Extraction: 10 min Analysis: 10 min	
		Dade Dimertest Latex Assay	0.025 µL dried stain	Analysis: 10 min	
		Lateral flow test, OneStep D-dimer RapidCard InstaTest	0.003125 µL dried stain	Analysis: 10 min	
Saliva	α-amylase	Lateral flow strip, RSID	1 µL saliva	Extraction: 50 min Analysis: 10 min	[26,30,33]
		Lateral flow strip, Seratec Amylase test & CS	1/1000 (50 mIU/mL)	Extraction: 10 min Analysis: 10 min	
Semen	Semenogelin	Lateral flow strip, RSID	1 µL semen	Extraction: 50 min Analysis: 10 min	[35,36,37,38,39,40,41,49]
	Prostate specific antigen (PSA)	Lateral flow strip, Seratec, PSA rapid test (Atlantic International Medical), Rapid PSA (Health Tech International)	0.5–1 ng/mL	Extraction: 50 min Analysis:10–15 min	
		Lateral flow strip, Bluestar, Identi-PSA	4 ng/mL	Extraction: 10 min Analysis: 10 min	
		Lateral flow strip, Onestep ABAcard PSA test	4 ng/mL	Analysis: 10 min	
Urine	Tamm-Horsfall protein	Lateral flow strip, RSID	10 µL urine	Analysis: 15 min	[42]
Biohazard	*B. anthracis* (anthrax)	Lateral flow strip, combined with NASBA	1.5 fmol	4 h	[44,45]
	*B. anthracis*, *F. tularensis* (tularemia), *Y. pestis* (plague)	Lateral flow dipstick, combined with:			[46]
		LAMP	100–1000 genome copies	Reaction: 60 min Analysis: 15 min	
		tHDA	100–1000 genome copies	Reaction: 30 min Analysis: 15 min	
		RPA	100–1000 genome copies	Reaction: 90 min Analysis: 15 min	
Donor profiling information	Sex typing	Lateral flow stip, combined with LAMP	10 pg	Reaction: 30 min Analysis: 30 min	[47]
Opioid	*M. speciosa*	Lateral flow strip PCR	0.01 pg	Reaction: <45 min Analysis: 10 min	[48]

## 4. Lateral Flow Assays for Chemical Forensic Applications

In the following subsections, the use of LFAs for illicit drugs analysis, biowarfare and detection of explosives is discussed. At the end, in Table 2 a summary is given of which LFA test(s) can detect which (classes of) analyte(s), including time-to-result and LOD.

### 4.1. Illicit Drugs

Nowadays many techniques are available for the analysis of illicit drugs, such as gas chromatography-mass spectrometry (GC-MS), liquid chromatography-mass spectrometry (LC-MS) and Raman spectrometry. However, these conventional analytical laboratory techniques are relatively bulky and require trained operators. The use of indicative tests is a valuable, cheap and time-consuming alternative [50]. 

For an indication of the presence or absence of illicit drugs in a sample usually presumptive colorimetric tests are used. These tests are used at the crime scene as rapid screening methods, but are also used in the lab for this purpose. Examples of such color tests are the Marquis test (morphine, 3,4-methylenedioxymethamphetamine (MDMA) and (met)amphetamine), the Scott test (cocaine), Simon’s test (methamphetamine and MDMA) and the Mandelin reagent (among others, ketamine, cocaine, MDMA and morphine). The exact structure of the complexes that are generated upon the reaction of the test reagents and the illicit drugs are in many cases not known [51,52]. 

As reported by Noviana et al. [53] tests based on a microfluidic paper-based analytical device (µPAD) are nowadays routinely used for the analysis of illicit drugs. Musile et al. [54] made a µPAD in which the unknown sample is split up into six different lanes, which can detect different types of compounds based on conventional color tests within 5 min. With this test multiplexed detection is possible of various (classes of) illicit drugs, such as cocaine, ketamine and morphine. 

Jang et al. [55] developed a drug kit to screen amphetamine-type stimulant drugs within beverages. Electrospinning was used to make the nanofiber-based paper with 10,12-pentacosadiynoic acid (PCDA)-dopamine as sensor material, which changes from blue to red upon contact with amphetamines. The results can be read out by the unaided eye or by taking a picture with a smartphone camera for further analysis. Dias et al. [56] worked on a device for the detection of illicit drugs in suspicious drinks. Their µOPTO (microfluidic optoelectronic tongue) device (Figure 5) for indicative use responds to alkaloid drugs, such as cocaine, morphine and ephedrine. The time-frame of 20 min before a digital image can be acquired is indicated as a potential drawback of their system. 

Phenacetin, a commonly used cutting agent or adulterant of cocaine, can be detected with an office paper-based device developed by da Silva et al. [57]. This device is based on previous work, which involved a colorimetric device made from qualitative filter paper for the detection of procaine (also a cutting agent) in cocaine samples [58]. A limit-of-detection (LOD) of 0.9 µmol/L (linear range of 5–60 µmol/L) and 3.5 µg/mL (linear range of 0–64.53 µg/mL) was found for procaine and phenacetin, respectively [57,58]. 

Angelini et al. [59] tested commercially available lateral flow strips for the detection of fentanyl (derivatives). The LOD of the test strips for fentanyl and norfentanyl was found to be 0.25 µg/mL and 0.05 µg/mL, respectively. The strips could be scored after 5–10 min. With the commercial Cozart RapiScan System (CRS) (also known as the Drug Detection System) also illicit drugs in oral fluids can be detected, such as (meth)amphetamine, MDMA and 3,4-methylenedioxy-N-ethylamphetamine (MDEA). Wilson et al. [60] concluded that the CRS is suitable for screening purposes, although the sensitivity and selectivity are lower compared to GC-MS. A total of 121 oral fluid samples tested positive (methamphe-tamine and MDMA) with the CRS, whereas 230 samples contained amphetamines (MDEA), 3,4-methylenedioxyamphetamine (MDA), MDMA or amphetamine according to confirmatory GC-MS. 

For the detection of Δ^9^-tetrahydrocannabinol (THC), cocaine (by the detection of benzoylecgonine (BZE)), opiates (through the detection of morphine) and amphetamine in sweat, Hudson et al. [61] developed a lateral flow device that uses a fingerprint as input. With the so-called Drug Screening Cartridge (Figure 6) fingerprint sweat samples were collected and analysed within 10 min. For read-out, the Intelligent Fingerprinting Reader 1000 Unit was used. The cut-off values were 190, 90, 68 and 80 pg for THC, BZE, morphine and amphetamine, respectively, resulting in accuracy levels of 93–99% for the four types of illicit drugs.

Taranova et al. [62] designed a lateral flow test for the simultaneous detection of morphine, amphetamine, methamphetamine and BZE based on a microarray. The microarray consisted of 8 × 3–4 rows yielding 24 or 32 detection spots. Since Taranova et al. [62] used a competitive assay, the color intensity of the spots decreases (up to complete disappearance) for increasing concentrations of the analyte. The test strip was held for 2 min in the analyte solution, whereafter the test could be read out after 8 min, thus a time-to-result of 10 min. LODs obtained were in the range of 1.2–20.1 mg/mL for the four analytes. 

A lateral flow test for the detection of illicit drugs in saliva was investigated by Carrio et al. [63] based on the DrugCheck SalivaScan from Express Diagnostics Inc. This test can detect a wide variety of illicit drugs, among others amphetamine, cocaine, ketamine and opiates. A sponge is swept through the mouth several times to collect the saliva and subsequently inserted into the device. Although the device could be read out by the unaided eye, Carrio et al. [63] applied a light box and smartphone read-out. Upon using the latter, fast (10 min for positive results), reliable and sensitive results could be obtained. 

### 4.2. Biowarfare

The rapid and secure detection of biowarfare agents or biotoxins is critical for forensic investigators. For example, ricin, botulin and aflatoxin can be used as potential agents of warfare or terrorist attack. Such agents and toxins are usually detected by an antibody-antigen reaction in an LFA [64]. 

Shyu et al. [65] developed a lateral flow immunoassay for the detection of ricin, of which the working principle is shown in Figure 7. Within 10 min down to 50 ng/mL ricin in phosphate-buffered saline (PBS) could be analyzed. When silver enhancement was used to increase the sensitivity of the assay down to 100 pg/mL could be detected [65]. The same research group also developed an assay for the detection of botulinum neurotoxin type B, the most poisonous toxin in the world. Concentrations down to 50 ng/mL and 50 pg/mL could be detected with a time-to-result of less than 10 min with the test and the silver enhanced test, respectively [66]. 

Hodge et al. [67] tested commercially available test strips for ricin. The strips could be read out by the unaided eye or by the use of the Rapid BioAlert Reader. For both 3 and 6 ng ricin the average time-to-read was found to be 19 min. The LOD for this assay was about 0.54 ng, which is equal to about 3.6 ng/mL. 

*B. anthracis*, the causative agent of anthrax, can be detected with a LFA based on antibody-antigen detection. Wang et al. [68] developed a LFA by which 400 pure spores could be detected within 30 min. As detection probe super-paramagnetic particles, a method to improve the sensitivity of the LFA, were used. 

Aflatoxin B1 (AFB1) can be detected by a dipstick type of LFA as developed by Shim et al. [64]. The detection principle is based on an aptamer and DNA probes, with a Cy5 coupled dye for fluorescent read out. The LOD was found to be 0.1 ng/mL AFB1 in buffer and 0.3 ng/g AFB1 for corn samples. The time-to-result of this LFA is 30 min. 

### 4.3. Explosives

When explosives are involved at a crime scene it is of utmost importance to collect fast information about the unknown substances. Since conventional on-site instrumental techniques, such as ion mobility spectrometry and infrared spectroscopy, can be relatively large and not easily operatable by, e.g., military or law enforcement, the use of (indicative) paper-based lateral flow devices has emerged [69]. 

Lateral flow assays for the detection of explosives are mainly based on either a colorimetric reaction or an antibody-antigen reaction. For the latter modified antibodies are present on the lateral flow strip that can bind the specific target molecule; a type of explosive in this case [70,71]. Similar to illicit drug analysis, explosive compounds can also be detected via a colorimetric reaction. Examples of these color tests are the Nessler reagent (2,4,6-trinitrotuluene (TNT), ammonium ions), Peroxides reagent (peroxides), Chloride reagent (chloride) and Griess reagents (cyclotrimethylenetrinitramine (RDX), pentaery-thritol tetranitrate (PETN), nitrate ions) [72]. 

TNT is one of the most used explosives. Romolo et al. [73] tested three immunochemical assays for the detection of TNT: (1) an indirect competitive ELISA with chemiluminescent detection (CL-ELISA), (2) a colorimetric lateral flow immunoassay (LFIA) and (3) a chemiluminescent lateral flow immunoassay (CL-LFIA). The first test was a laboratory test based on a microplate setup for comparison with the two LFIA tests. For the CL-ELISA trinitrobenzene and ovalbumin conjugates were used, whereby horseradish peroxidase (HRP) and luminol were used for detection, for the CL-LFIA anti-TNT antibodies were used with HRP as detection molecule, and for the LFIA test anti-TNT antibodies were implemented with colloidal gold as detection molecule. The colorimetric test had a limit-of-detection (LOD) of 1 µg/mL TNT. For the CL-LFIA the LOD was found to be 0.05 µg/mL (a conventional ELISA test is more sensitive with a LOD of 0.4 ng/mL). The time-to-result for the LFIA tests was 15 min. Also, Mirasoli et al. [74] developed a CL-LFIA for the detection of TNT, with a quantitative assay (antibody-antigen based, with anti-TNT antibodies and HRP as the detection molecule). Read-out was carried out with a CCD camera and with this setup a LOD of 0.2 µg/mL was obtained within 13 min. 

Chaboud et al. [75] developed a µPAD (Figure 8) for the colorimetric detection of metallic salts (lead, barium, antimony, iron, aluminum and zinc) present in primer residues and pyrotechnic low explosive devices. LODs of the metallic components were in the range of 0.025–0.4 µg, and the results could be observed with the unaided eye within 10 min. This device is based on earlier research on the development of a µPAD to detect inorganic explosives (e.g., flash powder and ammonium nitrate) and a µPAD to detect military explosives (e.g., TNT and urea nitrate). The first device was able to detect, among others, nitrate, nitrite and ammonium in combination with deionized water as solvent. TNT, RDX, hydrogen peroxide and urea nitrate could be detected by the second µPAD using 50%/50% acetone/water as solvent. Both devices utilized a colorimetric reaction of the explosive with the spotted reagent(s) with time-to-result within 5 min [76]. Taudte et al. [77] also applied a µPAD for the detection of various organic explosives (e.g., TNT, RDX and nitrobenzene) by using pyrene as fluorophore, which quenches in the presence of explosives. A prototype read-out system was developed and 0.1–0.9 µg of explosive material could be detected.

**Table 2 sensors-23-06201-t002:** Overview of characteristics of LFAs for illicit drugs, biowarfare and explosives.

Class of Analyte	Analyte(s) That Can Be Detected	Type of Test	LOD	Time-to-Result	Ref.
Illicit drugs	cocaine, codeine, thebaine, amphetamine, ephedrine, morphine, ketamine, MDMA and methamphetamine	Colorimetric µPAD	1.2 to 8.7 µg (MDQ)	<5 min	[54]
	amphetamine-type stimulants (in beverages)	Colorimetric nanofiber paper sensor	0.3 μg/μL (DSK-1) and 0.8 μg/μL (DSK-2)	immediate	[55]
	scopolamine, atropine, caffeine, cocaine, morphine, ephedrine, alprazolam and dipyrone	Colorimetric µOPTO	n.d.	20 min	[56]
	phenacetin (in seized cocaine)	Colorimetric paper device	3.5 μg/mL	1 min	[57]
	procaine (in seized cocaine)	Colorimetric paper device	0.9 µmol/L	n.d.	[58]
	fentanyl and norfentanyl	Rapid Response Fentanyl Test Strips *	0.25 µg/mL (fentanyl) and 0.05 µg/mL (norfentanyl)	5–10 min	[59]
	ampthetamines	Cozart RapiScan System *	1–5 ng/mL	5 min	[60]
	THC, opiates, cocaine and amphetamine (in sweat)	Drug Screening Cartridge *	68–190 pg	<10 min	[61]
	morphine, amphetamine, methamphetamine and BZE	Lateral flow strips (competitive assay)	1.2–20.1 mg/mL	10 min	[62]
	amphetamine, ketamine, cocaine, methamphetamine, opiates, marijuana and alcohol (in saliva)	DrugCheck SalivaScan *	5–50 ng/mL (cut-off values)	10 min	[63]
Biowarfare	recin	Immunoassay	50 ng/mL (with silver enhancement 100 pg/mL)	<10 min	[65]
	botulinum neurotoxin type B	Immunoassay	50 ng/mL (with silver enhancement 50 pg/mL)	<10 min	[65]
	recin	BioThreat Alert Test Strips *	3.6 ng/mL	<20 min	[67]
	*B. anthracis*	Immunoassay	400 pure spores	<30 min	[68]
	aflatoxin B1	Dipstick aptamer (competitive assay)	0.1 ng/mL	30 min	[64]
Explosives	TNT	LFIA and CL-LFIA	1 µg/mL (LFIA) and 0.05 µg/mL (CL-LFIA)	15 min	[73]
	TNT	CL-LFIA	0.2 µg/mL	13 min	[74]
	metallic salts	Colorimetric µPAD	0.025–0.4 µg	<10 min	[75]
	inorganic and military explosives	Colorimetric µPAD (2 types)	0.39–19.8 µg	<5 min	[76]
	organic explosives	Colorimetric µPAD	0.1–0.9 µg	n.d.	[77]

MDQ = minimum quantity detectable, DSK = drug screening kit, * = commercially available, n.d. = not determined.

## 5. Discussion

In this review, we have summarized the current applications of using LFAs for various forensic applications. In the past years, there have been numerous developments and improvements made regarding the use of LFAs within forensics. However, the majority of these developments have not led to a large scale of commercialized LFAs, except for body fluid identification and drug detection. Most of the developed LFAs are still in the conceptualization phase and need further validation and optimization to be feasible for the transition to the forensic field. In this section, the reasons for this lack of commercialization will be discussed in more detail. 

LFAs that are currently available for BFI for forensic practice show promising results with regard to sensitivity and specificity. However, care should be taken when mixtures of samples, or relatively rare body fluids, such as cerumen or breast milk are encountered at the crime scene, which have not been included in the validation studies so far (here ‘care’ means that it is important that when interpreting the results, the users should be aware that the test has not been validated with these relatively rare body fluids, meaning that it is not known whether such samples will provide (a) false-positive result(s) with the LFA). Additionally, the health status of an individual might also influence the amount of biomarker present in a certain body fluid, leading to possible false-positive or false-negative results. Laboratory testing is therefore still required to confirm the presence of a certain body fluid. To increase the specificity of the LFAs, multiplex analysis of more than one body fluid specific marker will be helpful. For forensic practice, the ability to analyze various body fluids at once will be of added value, since it reduces the risk of an inappropriately chosen body fluid assay. 

Although there are several manufacturers of commercial LFAs for BFI, for other biological forensic applications there are no LFAs on the market yet. For the detection of nucleic acid sequences of, e.g., pathogens or toxins various academic research groups investigated possibilities. Since an amplification reaction is necessary, these tests are rather time consuming and require external equipment (e.g., a hot plate), which makes them unsuitable as tests on the crime scene. 

LFAs for the analysis of illicit drugs are available on the market and also deployed at the scene of interest. There are also kits available for personal use, e.g., to test for THC, amphetamine and methamphetamine in urine or saliva. Although LFAs for biowarfare and explosives detection evolved by researchers, commercialization of such LFAs is lacking (for unknown reasons). 

Overall, as follows from Section 3 and Section 4, a variety of LFAs is reported for BFI, and for the detection of DNA, illicit drugs, biowarfare and explosives. However, despite their advantages, the use of LFAs within forensics is still rather limited, which may be related to the time-to-result of reported LFAs. In order to come to broadly applicable LFAs at a crime scene a time-to-result within 10 min is required. The majority of LFAs for BFI and illicit drugs analysis (as well as explosives and biowarfare) meet this requirement, however, LFAs for DNA testing do not (yet). The latter tests take 45 min up to several hours, which makes their use at the scene of interest inoperable. Development of low-cost LFAs that can detect a specific (class of) analytes within 10 min might boost the use of LFAs within forensics, in particular if such LFAs are multiplex, i.e., capable of detecting more than one (class of) analytes.

For most reported LFAs read-out is done by the human eye, which avoids the use of large and complex read-out equipment. However, a disadvantage of readout and/or interpretation by the human eye is that the skills and experience of the operator play a role, as well as the amount of environment light (well-lit vs. poor light conditions) [63]. Developments leading to less subjective read-out will be beneficial with regard to the use of more LFAs in the forensic field. 

Although the indicative aspect of most LFAs can be seen as a disadvantage (no ‘real evidence’ can be obtained from indicative LFAs), it may be clear that the initial information that can be gained with these LFAs can help/direct the initial phases of an investigation as well as speed up the (start of the) investigation trajectory. Nevertheless, the use of indicative LFAs implies that confirmation by using conventional methods and techniques is required [63]—for example DNA analysis after the use of LFAs [78]—which seems another aspect that hampers/delays broad use of these LFAs in forensics.

Finally, although the costs for commercially available LFAs are relatively low (i.e., a few euro/dollars to a few tens of euro/dollars), other costs that need to be considered in case of use of LFAs at crime scenes are, for example, training of personnel. However, the expected cost for this training will depend on the application. For instance, for body fluid identification, the LFAs are user-friendly and easy to use. A short instructional video will be sufficient to create the background knowledge to handle such LFA at the crime scene. In contrast, for the application of analyzing DNA samples, different skills are needed, which also require more elaborated training.

## 6. Conclusions

The use of LFAs for forensic applications is reviewed. Due to their advantages—viz. ease-of-use, relative low-cost as well as sufficient sensitivity and sensitivity—LFAs are nowadays mainly used for screening purposes for BFI, illicit drug analysis and detection of explosives and biowarfare. In order to become applicable for more forensic applications, the time-to-result of LFAs has to be reduced: an outcome within 10 min is required at a scene of interest (e.g., a crime scene). Moreover, it seems essential that such LFAs are able to perform multiplex analyses. Overall, LFAs have been partially implemented in forensic case work and can be of additional use to extract important information directly at the crime scene. However, more research is needed to improve current LFAs and make the transition to commercialization and implementation.

## Figures and Tables

**Figure 1 sensors-23-06201-f001:**
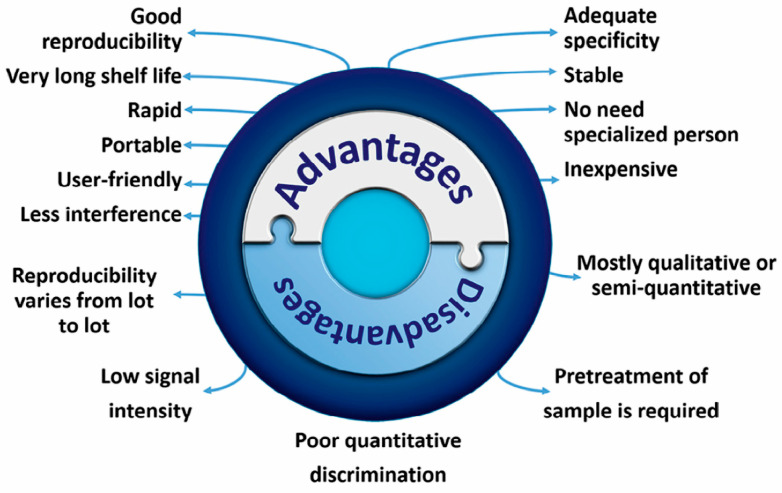
Advantages and disadvantages of lateral flow assays. “Reprinted from [17], with permission from Elsevier”.

**Figure 2 sensors-23-06201-f002:**
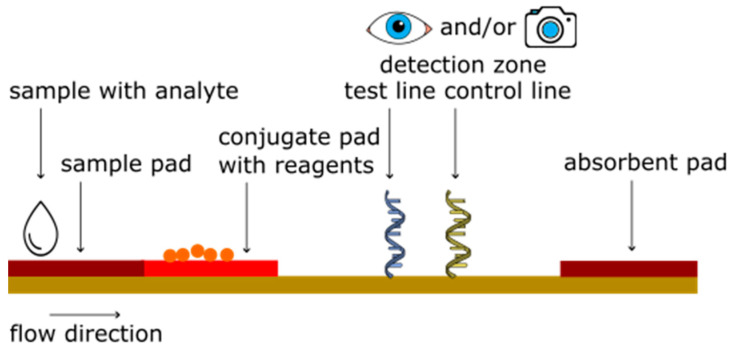
Overview of the various parts and working principle of a lateral flow assay.

**Figure 3 sensors-23-06201-f003:**
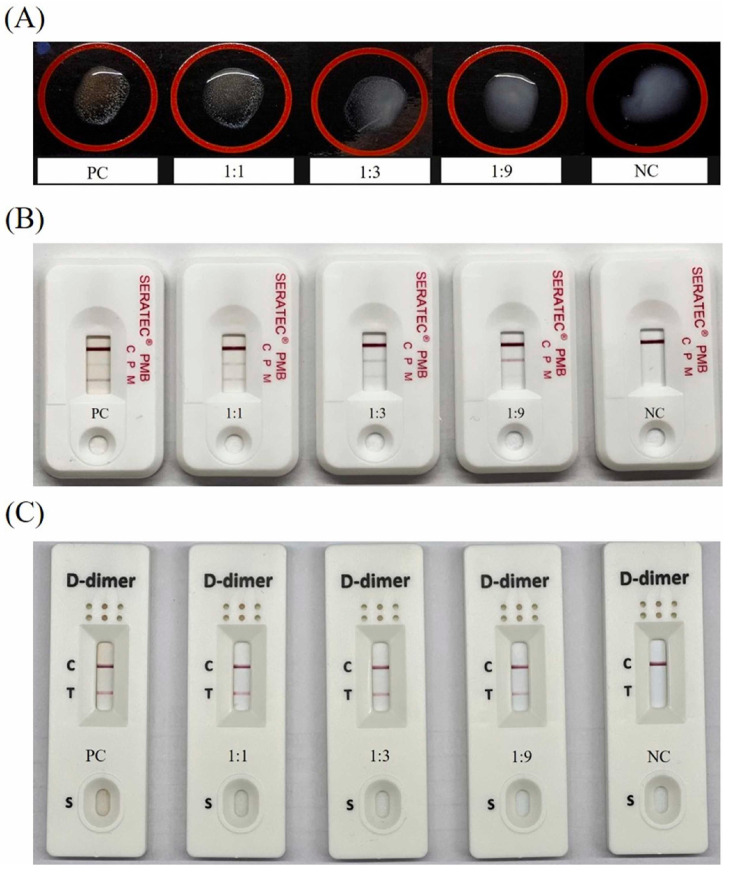
Results of three commercial kits for menstrual blood samples mixed with PBS at three different ratios. (**A**) Dade Dimertest Latex Assay, (**B**) SERATEC PMB test and (**C**) OneStep D-dimer RapidCard InstaTest. NC = negative control (PBS), PC = positive control (pure menstrual blood). “Reprinted from [26], with permission from Elsevier”.

**Figure 4 sensors-23-06201-f004:**
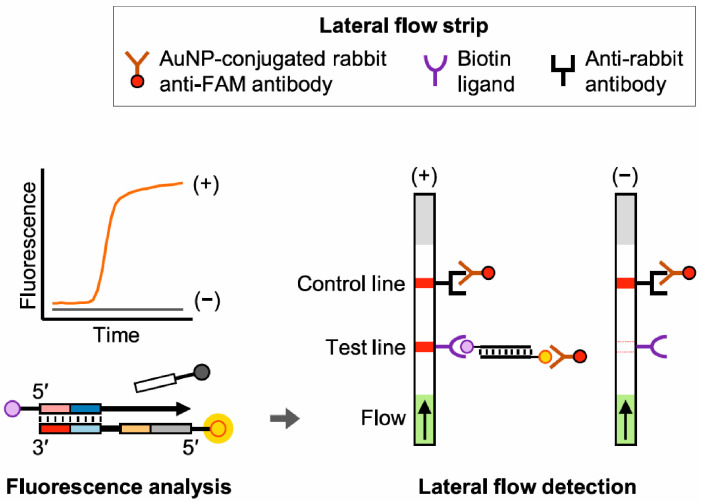
Schematic representation of the fluorescence analysis and lateral flow detection. The LAMP assay produces a fluorescent signal and simultaneously generates a dual-labelled amplicon with FAM and biotin. By capturing the amplicon on the lateral flow strip a visible band is generated. “Reprinted from [47], with permission from Elsevier”.

**Figure 5 sensors-23-06201-f005:**
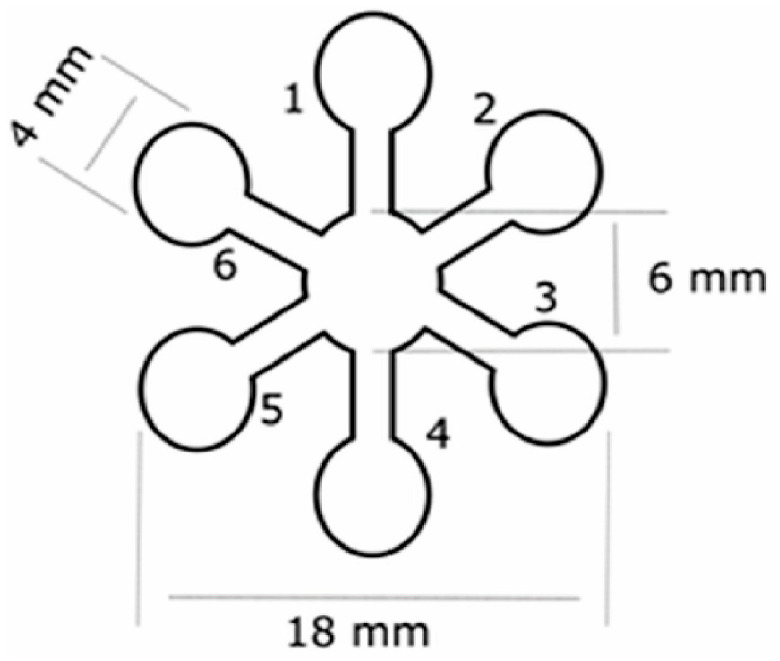
Schematic overview of the µOPTO device with 6 spots for various reagents (1: tetraphenylporphyrin (ZnTPP), 2: methyl orange, 3: bromocresol green, 4: iodoplatinate, 5: Dragendorff’s, 6: Chen’s). “Reprinted from [56], with permission from Elsevier”.

**Figure 6 sensors-23-06201-f006:**
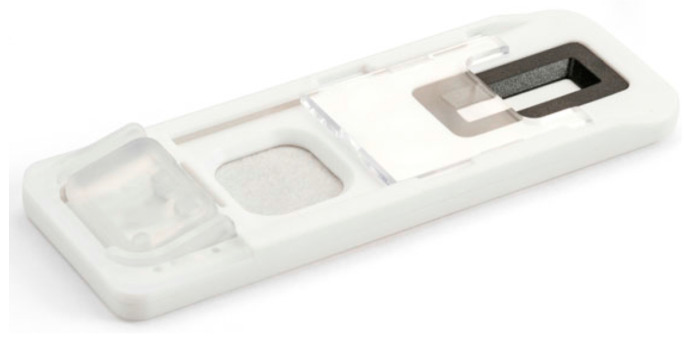
Image of the Drug Screening Cartridge. “Reprinted from [61], with permission from Oxford University Press”.

**Figure 7 sensors-23-06201-f007:**
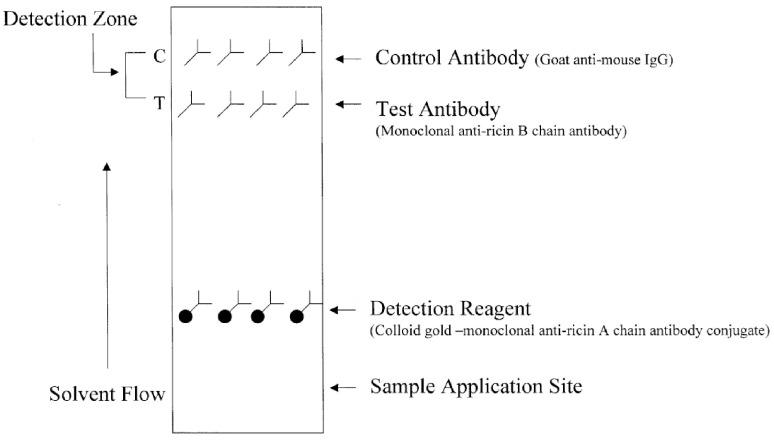
Schematic description of an immunochromatic device for the detection of ricin. “Reprinted from [65], with permission from Elsevier”.

**Figure 8 sensors-23-06201-f008:**
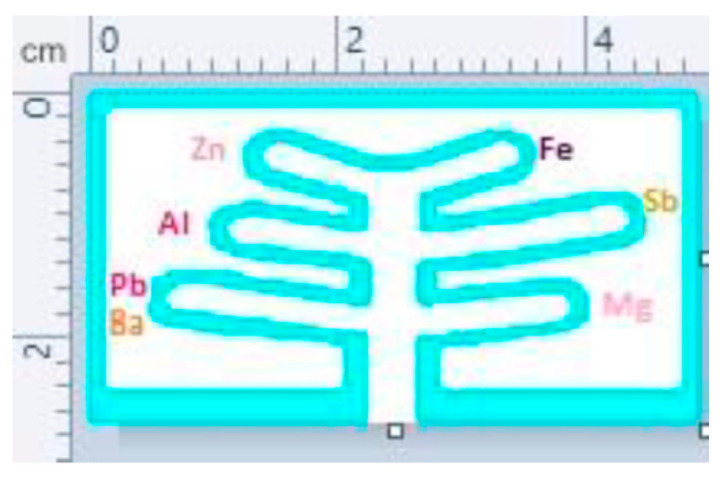
A multiplexed µPAD for the detection of various metallic salts. Each lane is labelled with the colour at which each analyte should appear in the case of a positive reaction. “Reprinted from [75], with permission from Elsevier”.

## Data Availability

Not applicable.

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
