# Peer review of "Use of Lateral Flow Assays in Forensics"

_sensors, 2023, doi:10.3390/s23136201_

Round 1

Reviewer 1 Report

The manuscript describes and discuss the most recent developments in lateral flow assays related to the forensic field. The paper is well written, and the information is useful, so I consider that it can be published with minor corrections.

Suggestions and comments:

Line 153. Please amend semenogelin units.

It is suggested to incorporate a table in which the main characteristics of the reported devices should be summarized: sample type, detection limit, analysis time, detection mechanism, etc.

Author Response

See attachment for our responses to the points of Referee #1

Reviewer 2 Report

This paper gives a comprehensive review of LFAs in the forensic sciences.  The manuscript then provides a discussion on why LFAs are not widely used in various forensic science disciplines. 

The paper gives a thorough overview of LFAs.  By comparison the discussion is very short and still leaves the reader with a number of questions:

1).  The authors say "care should be taken" when mixtures of samples are encountered.  At crime scenes, mixed samples are encountered frequently.  How should this be address with LFAs?  What "care" should be exercised.

2).  The authors state that validation studies have not be undertaken.  Why is this the case?  Few, if any, test protocols can be utilized in the forensic sciences without validation studies.

3).  The authors make a short statement about LFAs needing confirmation testing by traditional forensic science testing.  Which is true, but that is a major factor in implementation.  It is time and costs, which then have to be repeated later.  If that is the case, why bother implementation at the scene?  The authors would make a stronger case by pointing out the initial information that can be gained, which could help direct the initial phases of the investigation.  This could be justification for the implementation of the extra time and expense for LFA implementation.

4).  The authors do not mention the training needed for the implementation of LFAs at the crime scene.  As simple as they may be, awareness and use training is needed, and that is also an additional cost.

It seems this paper could benefit from a revision of the discussion to make it more expansive and inclusive of the difficulties, expense, personnel costs, and training needed for the limited benefits of LFAs at the scene.  And include a broader detail of those benefits.

Author Response

See attachment for our responses to the points of Referee #2

Reviewer 3 Report

The Authors present an article regarding the application of Flow assays in forensic science.

The topic is definitely of interest for the forensic community because it explore a particular reality of the research and it might be helpful in moving it forward to considering the important application that this activity can have in the forensic field as in the  analysis of the crime scene.

Furthermore, the value of this manuscript is also due to the review that the Authors made with the papers published in scientific literature.
The manuscript is well organized and corrected in the methodology of the research, the writing is clear and concise, limits and advantages are described.

The only suggestion concerns the addition of a graphical abstract which would help the reader to immediately focus on the content of the manuscript.

The work should definitely be published.

Author Response

See attachment for our responses to the points of Referee #3

Reviewer 4 Report

This is a very interesting article regarding the uses of LFA's in forensic practice, synthesising the most important applications that have been developed in this area.

Some minor comments:

- I would not start the abstract with the use of LFAs in the COVID-19 pandemic. They have been  widely used before. This seems more like a "catchy" introduction.

- when discussing tests, it is very important to mention specificity and sensibility (if known)

- maybe it would be useful to synthesise the main LFA's that are commercially available, for each type, in a table, showing the uses, Sb, Sp, producer and references showing their practical applications as published in scientific papers/conferences.

- the authors emphasise the low cost of these LFA's in the abstract, but fail to mention costs anywhere else in the article. It would also be interesting to present a cost-benefit analysis of these LFAs compared to similar technologies. 

Author Response

See attachment for our responses to the points of Referee #4
